# Weather Trumps Festivity? More Cardiovascular Disease Events Occur in Winter than in December Holidays in Queensland, Australia

**DOI:** 10.3390/ijerph181910158

**Published:** 2021-09-27

**Authors:** Clifford Afoakwah, Son Nghiem, Paul Scuffham, Simon Stewart, Joshua Byrnes

**Affiliations:** 1Centre for Applied Health Economics, Griffith University, Nathan, QLD 4111, Australia; s.nghiem@griffith.edu.au (S.N.); p.scuffham@griffith.edu.au (P.S.); j.byrnes@griffith.edu.au (J.B.); 2Menzies Health Institute Queensland, Griffith University, Southport, QLD 4215, Australia; 3Centre for Cardiopulmonary Health, Torrens University Australia, Adelaide, SA 5000, Australia; simon.stewart@Torrens.edu.au

**Keywords:** cardiovascular disease, winter, Christmas holidays, hospitalisations, deaths

## Abstract

*Objective:* Cardiovascular disease (CVD) is the leading cause of hospitalisations and deaths in Australia. This study estimates the excess CVD hospitalisations and deaths across seasons and during the December holidays in Queensland, Australia. *Methods:* The study uses retrospective, longitudinal, population-based cohort data from Queensland, Australia from January 2010 to December 2015. The outcomes were hospitalisations and deaths categorised as CVD-related. CVD events were grouped according to when they occurred in the calendar year. Excess hospitalisations and deaths were estimated using the multivariate ordinary least squares method after adjusting for confounding effects. *Results:* More CVD hospitalisations and deaths occurred in winter than in summer, with 7811 (CI: 1353, 14,270; *p* < 0.01) excess hospitalisations and 774 (CI: 35, 1513; *p* < 0.01) deaths compared to summer. During the coldest month (July), there was an excess of 42 hospitalisations and 7 deaths per 1000 patients. Fewer CVD hospitalisations (−20 (CI: −29, −9; *p* < 0.01)) occurred during the December holidays than any other period during the calendar year. Non-CVD events were mostly not statistically significant different between periods. *Conclusion:* Most CVD events in Queensland occurred in winter rather than during the December holidays. Potentially cost-effective initiatives should be explored such as encouraging patients with CVD conditions to wear warmer clothes during cold temperatures and/or insulating the homes of CVD patients who cannot otherwise afford to.

## 1. Introduction

Seasonal variations in cardiovascular disease (CVD) events have been established in the Northern hemisphere. Such variations, often attributable to extreme low and high temperatures during winter and summer, respectively, can lead to CVD events, reaching their peak in winter [1,2,3]. The southern hemisphere is no exception to such seasonality in CVD events [4,5,6]. In the northern hemisphere, in addition to cold temperatures, winter is characterised by Christmas and New Years’ festivity in December, which substantially increases the incidence of CVD events more than any comparable period in the calendar. Contrarily, the December holidays (Christmas and New Year festivities) coincide with summer in the southern hemisphere.

In countries where long (essential) holidays such as Christmas Day, and New Year’s Day holiday are celebrated, CVD mortality has been shown to peak during these periods [7,8,9,10]. Studies on Christmas-related CVD events have largely focused on the northern hemisphere and have shown consistent evidence that mortality peaks during the Christmas holidays [8,9,10]. The excess CVD mortality during these periods is attributed to the changes in people’s physical environment such as visiting friends and relatives, large intake of alcohol and unhealthy food, and extreme physical activities [11,12,13]. Evidence on the association between festivity holiday seasons and CVD events in the southern hemisphere is scant. Knight and Schilling [7] found little (~4 deaths per year) but significant evidence of excess deaths during the Christmas period. Although, New Zealand does have relatively lower temperatures during summer compared to other countries in the southern hemisphere.

Queensland in Australia, on the other hand, has a warm and tropical climate, which is Humid Sub-Tropical. During summer, it has average summer temperatures of 25.5 °C–32.3 °C, making it one of the hottest states in Australia. Besides the hot temperature, Queensland is the hotspot for CVD in Australia, with estimated heart failure-related admissions of 14,000 in 2019 [14], providing an ideal context to re-examine the linkages between seasonality, December holidays and CVD events in the southern hemisphere. The objective of this study was to estimate excess CVD hospitalisations and deaths across seasons and during the December holiday period using retrospective, longitudinal, population-based cohort data from Queensland, Australia.

## 2. Methods

### 2.1. Data

Data for the study are sourced from the Queensland Cardiovascular Data linkage (QCard) [15]. QCard is linked longitudinal administrative data sourced from different administrative databases. This study utilises two of the many databases: Queensland Hospital Admitted Patient Data Collection (QHAPDC) and Registrar General Deaths Database (RG). The population includes 75,829 individuals who were first admitted as an inpatient with a diagnosis of CVD to one of the 247 health care facilities in Queensland, Australia in 2010. Data on subsequent hospitalisations until December 2015 were collected, resulting in a total of 592,296 hospitalisations. Mortality data on these cohorts were obtained from the Registrar General Deaths Database (RG). Detailed description of the QCard data is reported elsewhere [15].

### 2.2. Outcomes

Two main CVD events (outcomes) were considered in this study: hospitalisation and death. Hospitalisations were categorised into two groups: whether CVD was the primary diagnosis (CVD-related) or secondary diagnosis (non-CVD-related). Similarly, deaths were categorised as whether they were CVD-related or non-CVD-related. The Registrar General Deaths Database includes information on the cause of death. We, therefore, compared seasonality and the December holiday period for CVD-related versus non-CVD-related events.

### 2.3. Statistical Analyses

CVD events (hospitalisations and deaths) were grouped according to whether they occurred in winter (the months of June, July and August), spring (September, October and November), summer (December, January and February) or autumn (March, April and May) to examine broad seasonal patterns of CVD events. The December (Christmas) holiday period was defined as a 14-day window starting from 24th December to 6th January [7]. Two comparator periods were used: 14 days pre and post the Christmas holiday period, and any other day besides the pre- and post-Christmas period. Excess hospitalisations and deaths were estimated after adjusting for sex, age at death, indigenous status and year trend. This approach allows unobservable characteristics that vary across time to be isolated from the observed seasonal and Christmas holiday patterns in CVD events. In the analysis for December holiday CVD events, we also control for the month of events dummies. The number of fewer/excess CVD events per period was then estimated using the ordinary least squares (OLS) method. In the analyses for seasonality, summer is used as the reference season while January, the hottest month, was used as the reference month for a month-by-month analysis of CVD events. Additionally, all other days was the comparison period for the Christmas holiday analysis. All analyses were performed using STATA v16, and statistical significance was accepted at two-sided *p*-values less than 0.05

## 3. Results

### 3.1. Descriptive Results

Table 1 shows the demographic characteristics of CVD and non-CVD-related events analysed in this study. A total of 592,296 hospitalisations were analysed in this study, of which 181,890 (30.7%) were CVD-related and the remaining 410,406 (69.3) were non-CVD-related. Additionally, 14,035 (18.5) patients died during the study period, of which 7366 (52.4%) were related to CVD and the remaining 6669 (47.6%) were non-CVD-related. Males accounted for 53.5% of all hospitalisations and 52.4% of all-cause deaths. Although the proportion of males did not substantially differ between CVD-related and non-CVD-related hospitalisations, the proportion of males who died from non-CVD (54.8%) was significantly higher than those who died from CVD (50.1%). Indigenous patients constituted 2.8% of all hospitalisations and 2.2% of all-cause deaths. Most indigenous patients were hospitalised with CVD as secondary diagnosis (3.0%) and died from non-CVD (2.6%). Additionally, while the majority of hospitalisations among those aged 17–74 years were non-CVD-related, most of those aged above 74 years were hospitalised with CVD as the primary diagnosis (36.5%). Consistently, patients aged above 74 died predominantly from CVD (77.9%) compared to those aged 74 and below.

### 3.2. Seasonality in CVD Hospitalisations and Deaths

Figure 1 shows seasonal results of CVD and non-CVD events. It shows that CVD hospitalisations reached their peak in winter with an average of 7811 (CI: 1353, 14,270; *p* < 0.001) excess hospitalisations over summer. Since the study uses population-based data, the total of 75,829 patients implies that per 1000 CVD patients, there were 103 excess hospitalisations during winter. Additionally, Figure 2 shows that each year, there was an excess of 774 (CI: 35, 1513; *p* < 0.01) CVD deaths during the winter season, which is equivalent to 10 deaths per 1000 patients. Intriguingly, the winter season was not the peak of mortality: CVD deaths peaked in spring with an excess of 1256 (CI: 467, 2045; *p* < 0.001) deaths per annum. Non-CVD hospitalisations and deaths did not show any significant seasonal pattern.

### 3.3. Monthly Variations in CVD and Non-CVD-Related Events

Figure 2 shows that on a month-by-month basis, there was an excess of 3229 (CI: 1381, 5076; *p* < 0.01) CVD-related hospitalisations during the coldest month of July over January (the hottest month). Additionally, CVD deaths significantly peak in July, with an excess of 552 (CI: 218, 886; *p* < 0.01) deaths over January. Equivalently, there was an excess of 42 hospitalisations and 7 deaths per 1000 patients during July. Interestingly, July was not the peak of CVD hospitalisation; rather August, at the end of the winter season, was the peak with an excess of 3481(2769, 5192; *p* < 0.01) hospitalisations over January, equivalent to 46 excess hospitalisations per 1000 patients.

### 3.4. CVD- and Non-CVD-Related Events during December Holidays

Finally, Figure 3 reports the number of excess CVD and non-CVD events during the Christmas holiday period. Figure 3 shows that there were 20 (CI: −29, −9; *p* < 0.01) fewer CVD hospitalisations during the December holiday period compared to any other period in the calendar year. Additionally, there were fewer CVD hospitalisations compared with 14 days before and after the holiday period. Although CVD deaths during the holidays were lower than all other days, these were not statistically significant. Non-CVD-related hospitalisations during the holidays were not significantly significant; however, non-CVD deaths were significant but lower than CVD deaths during the Christmas holiday period.

### 3.5. Sensitivity Analyses

Two sensitivity analyses were conducted. First, we explored the trends in the estimated Christmas-related CVD events. Figure A1 in Appendix A shows that Christmas CVD events were consistent with the main findings in Figure 3. Additionally, while CVD hospitalisations were reducing over time, CVD deaths were increasing among the cohort. Overtime, there would be excess CVD hospitalisations during the December holiday period but CVD deaths during the same period would still be insignificant.

In the second sensitivity analyses, we focused on the peak of the Christmas festivity; the Christmas holiday period was redefined as 25–27 December and compared with 25–27 of any other months (January to November) as well as all other days. Results from Figure A2 show that irrespective of how the Christmas holiday period is defined or the comparator period chosen, there are no excess Christmas-related CVD-related events; rather, there are significantly fewer CVD-related events.

## 4. Discussion

The Christmas holidays is arguably a risk factor for CVD events due to the stress, change in one’s physical environment and the high intake of unhealthy food and alcohol, which independently are risk factors for CVD events. This study has utilised large linkage longitudinal cohort data of patients with pre-existing CVD conditions to re-examine seasonality and Christmas-related CVD events from Australia, in the southern hemisphere, where Christmas does not coincide with the summer period.

The findings show that CVD hospitalisations peaked in winter, particularly the month of August. During the winter season, there was an excess of 103 hospitalisations and 10 deaths per 1000 patients. Although CVD deaths peaked during spring, excess deaths attributable to winter were 774 (equivalent of 10 per 1000 patients) over summer. Our estimates for excess winter deaths are higher than the 2.8 per 100 deaths found by Weerasinghe and MacIntyre [5] in New South Wales. One plausible reason for such difference is that while this study focuses on all types of CVD conditions, Weerasinghe and MacIntyre [5] focused on only coronary artery disease. Additionally, the high CVD deaths rate in Queensland compared to other Australian states [16] partly explains such high excess mortality in this study.

On a month-by-month basis, CVD hospitalisations peaked in August, while deaths peaked in July—the coldest month of the year. This suggests that the spike in CVD deaths in July is attenuated by relatively lower deaths in June and August. In July, there was an excess 43 hospitalisations and 7 deaths per 1000 patients, over the hottest month of January. There are increased environmental provocations such as cold temperatures, reduced daylight hours, and respiratory infections during winter that lead to CVD events [17,18]. This is because cold temperatures cause bronchoconstriction and suppresses mucociliary defence as well as other immunological reactions [19,20,21]. The increased risk of infections due to persistent cold temperatures can, therefore, trigger adverse health events if emergency or precautionary measures are lacking.

Finally, analysis of Christmas-related CVD events showed that there were fewer CVD hospitalisations compared to similar periods pre- and post-Christmas and all other days in the calendar year. Specifically, per 10,000 patients, there were ~3 excess hospitalisations compared to all other days in the calendar year. Although there were fewer CVD deaths over the Christmas holiday period compared with other days, this was not statistically significant.

The cost of the 7811 excess winter hospitalisations in Queensland is substantial. Each hospitalisation costs approximately AUD 2358 as estimated from the QCard data by Afoakwah and Nghiem [22]. Thus, the total hospital costs of excess CVD hospitalisations in winter are ~AUD 18.4 million to the health sector.

The only study in the southern hemisphere [7] that found evidence of excess CVD deaths during Christmas, equivalent to four deaths per year, was undertaken in New Zealand; their estimate was by far lower than excess Christmas deaths recorded in the northern hemisphere [9,10,11,23]. Although Weerasinghe and MacIntyre [5] did not directly investigate CVD events during the Christmas holiday period, the authors found that coronary artery death did not statistically increase before and after Christmas periods in New South Wales. Our findings on Christmas-related CVD events are contradictory to those found in the northern hemisphere. In Scandinavia, for example, Mohammad and Karlsson [10] and Moholdt and Afoakwah [6] showed that CVD mortality peaked during Christmas/New Year holidays, while similar evidence has been reported in the United States [8,11] and Canada [23]. This contradictory evidence can be attributed to differences in the season that coincides with the December/Christmas holiday. While, in the northern hemisphere, Christmas holidays occur in winter, in the southern hemisphere, particularly Queensland, Australia, Christmas holidays occur in summer.

The findings from our study add to the few studies on Christmas-related CVD events in the southern hemisphere. While our study is the first to analyse Christmas-related CVD hospitalisation in the southern hemisphere using longitudinal cohort data, the estimates of excess CVD events across seasons are unique, relative to previous studies. The findings of more CVD hospitalisation and deaths in winter than the Christmas holidays are particularly important for policy makers to minimise CVD events in a cost-effective manner.

Despite the strength of this study, its main weakness is the relatively shorter follow up period of the cohort. While we acknowledge that short time period may compromise long term predictions, the analytical approach used, which controls for year trend in CVD events, helps to minimise any potential biases.

## 5. Conclusions

This study has estimated the excess CVD-related events across seasons and during the December holiday period using retrospective, longitudinal, population-based cohort data from Queensland, Australia. Our key finding is that winter, rather than December holidays, is the major cause of CVD events. Government policies that seek to minimise CVD events should target the winter period, which is often characterised by cold temperatures, instead of the Christmas holiday period. These may include but are not limited to encouraging patients with CVD conditions to wear warmer clothes during cold temperatures and insulating the homes of CVD patients who cannot afford it. A more complementary policy approach can minimise the extremely high CVD events in winter.

## Figures and Tables

**Figure 1 ijerph-18-10158-f001:**
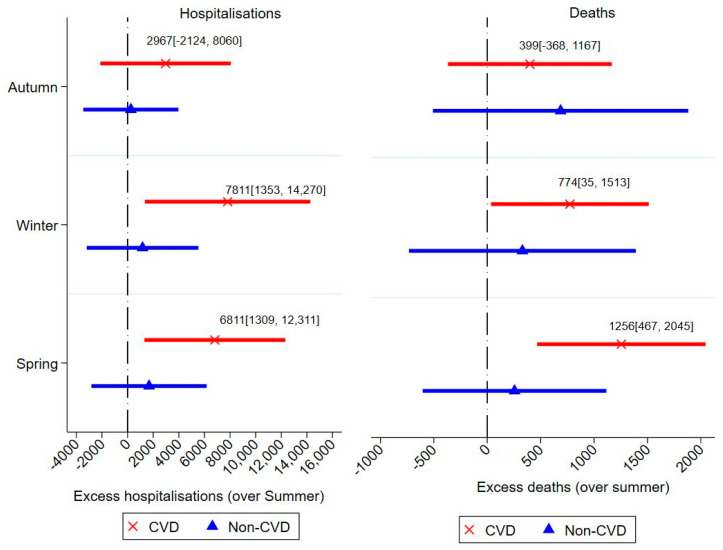
Excess CVD and Non-CVD events across seasons.

**Figure 2 ijerph-18-10158-f002:**
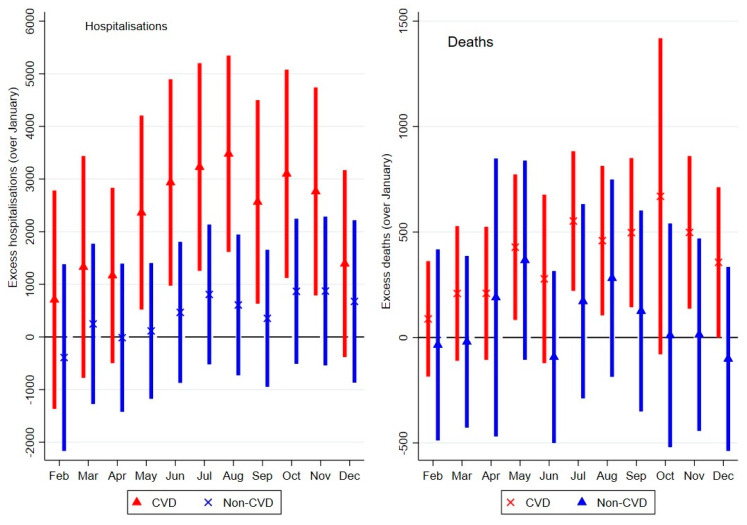
Excess monthly CVD- and non-CVD-related events.

**Figure 3 ijerph-18-10158-f003:**
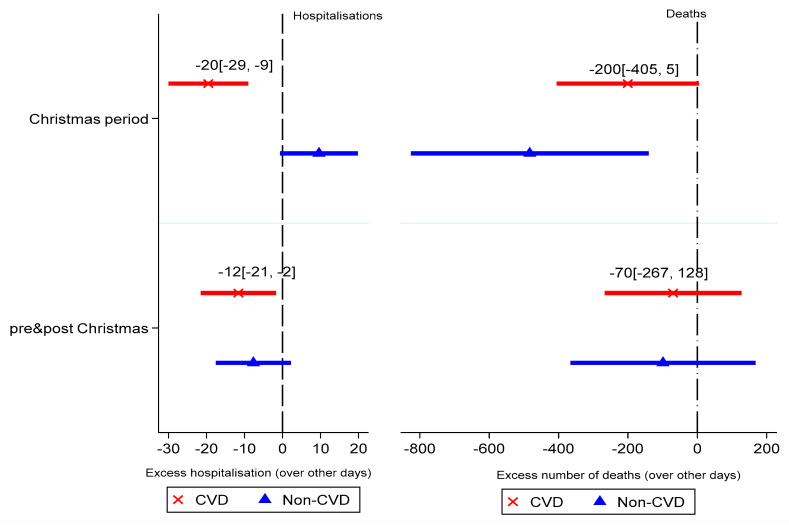
CVD- and non-CVD-related events during December holidays.

**Table 1 ijerph-18-10158-t001:** Demographic characteristics of CVD- and non-CVD-related events.

	Hospitalisations				Death			
	All-Cause *n* = 592,296	CVD *n* = 181,890	Non-CVD *n* = 410,406		All-Cause *n* = 14,035	CVD *n* = 7366	Non-CVD *n* = 6669	
	%	%	%	*p*-value	%	%	%	*p*-value
Male	53.5	53.6	53.4	*p* = 0.128	52.4	50.1	54.8	*p* < 0.01
Indigenous	2.8	2.5	3.0	*p* < 0.01	2.2	2.0	2.6	*p* < 0.01
Age								
17–44	11.2	10.1	11.4	*p* < 0.01	2.8	1.1	4.7	*p* < 0.01
45–64	31.5	30.2	32.0		16.1	9.6	23.3	
65–74	25.0	23.2	25.8		22.1	17.1	27.5	
>74	32.3	36.5	30.7		65.9	77.9	52.6	

## Data Availability

Restrictions apply to the availability of these data. Data were obtained from Queensland Health and the Australian Institute of Health and Welfare and stored at the Secure Unified Research Environment (SURE) at the SAX Institute, Australia. Access to this dataset is subject to ethical approval from all the data custodians.

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
