# Peer review of "Weather Trumps Festivity? More Cardiovascular Disease Events Occur in Winter than in December Holidays in Queensland, Australia"

_ijerph, 2021, doi:10.3390/ijerph181910158_

Round 1

Reviewer 1 Report

Overall I found this study interesting and convincing. Report should be improved a bit, though:

Table 1 reports percentages for categories, yet those are labelled 'Mean'

Base for % is the respective group/subgroup CVD/nonCVD. When comparing within groups like male or idifenous, the base should be that group. e.g. CVD was not the main cause of death among males (54.8%).  Either report the % of all males dying from nonVD causes or rephrase like The rate of males with non_CD death was higher than for CVD death...

There is an error in reporting on indegenous patients:

Most indigenous patients were hospitalised with CVD as secondary diagnosis (3.0%) and died from CVD (2.6%). It is nonCVD !

Figure 2 (not 3 as in line 131) could be transformed to a line graph to convey the message of seasonality more clearly.

Author Response

Reviewer 1

Overall I found this study interesting and convincing. Report should be improved a bit, though:

Table 1 reports percentages for categories, yet those are labelled 'Mean'

Response: Thanks for this comment. All variable reported in Table 1 are binary, so the means represent the proportion (percent) of the selected variables. However, we have changed the label to percentages and accordingly reported Table 1 as percentages instead of means. This is available on page 3.

Base for % is the respective group/subgroup CVD/nonCVD. When comparing within groups like male or idifenous, the base should be that group. e.g. CVD was not the main cause of death among males (54.8%).  Either report the % of all males dying from nonVD causes or rephrase like The rate of males with non_CD death was higher than for CVD death...

Response: We appreciate this comment and have rephrased this sentence as “Although the proportion of males did not substantially differ between CVD-related and non-CVD related hospitalisations, the proportion of males who died from non-CVD (54.8%) was significantly higher than those who died from CVD (50.1%)” on page 3 lines 110-112.

There is an error in reporting on indegenous patients:

Most indigenous patients were hospitalised with CVD as secondary diagnosis (3.0%) and died from CVD (2.6%). It is nonCVD !

Response: Thanks a lot for noting this. It has been corrected accordingly in line 114 on page 3

Figure 2 (not 3 as in line 131) could be transformed to a line graph to convey the message of seasonality more clearly.

Response: Thanks for this comment. We have redrawn Figure 2 in a vertical form making the seasonality message clearer than before. However, we could not impose the line graph in our coefficient plot.

Reviewer 2 Report

Very nice paper on the influence of seasons and Christmas holiday on CV outcomes (hospitalization and death). I commend the Authors for their job and my comments are mostly minor:

  • line 22, please rephrase the following sentence: "Fewer CVD events [-20 (CI: -29, -9; p<0.01)] CVD hospitalisations..."
  • line 194, ref. 17 is an appropriate self-citation
  • Discussion: I would describe how do you think winter season may increase the risk of CV outcomes (e.g. vasoactive effect of temperature). It would also be interesting to perform a sensitivity analysis based on patients income to understand if low income is associated with an excess mortality and hospitalization during winter but not during spring/autumn 

Author Response

Reviewer 2

Very nice paper on the influence of seasons and Christmas holiday on CV outcomes (hospitalization and death). I commend the Authors for their job and my comments are mostly minor:

  • line 22, please rephrase the following sentence: "Fewer CVD events [-20 (CI: -29, -9; p<0.01)] CVD hospitalisations..."

Response: Thanks for this. We have revised this sentence in line 23 as “Fewer CVD hospitalisations [-20 (CI: -29, -9; p<0.01)] occurred during the December holidays than any other period during the calendar year”

  • line 194, ref. 17 is an appropriate self-citation

Response: We have revised this on line 203

  • Discussion: I would describe how do you think winter season may increase the risk of CV outcomes (e.g. vasoactive effect of temperature).

Response: We have included some explanations on why winter season may increase the risk of CVD events on line 189-194 on page 7.

  • It would also be interesting to perform a sensitivity analysis based on patients income to understand if low income is associated with an excess mortality and hospitalization during winter but not during spring/autumn 

Response: Thanks for this comment. We agree with you that some sensitivity analyses on how the estimated winter-related CVD events varies among patients with different income level will be very useful, especially for policy purposes. However, we do not have data on individual income in our administrative data since hospitals do not collect information on the patient’s income.

Reviewer 3 Report

The authors investigated CVD hospitalisations and deaths across seasons and during the December holiday period using retrospective, longitudinal, population-based cohort data from Queensland, Australia (from January 2010 to December 2015). Overall, it's an interesting study with strong clinical relevance and significance. I only have a few comments.

  1. The authors should compare their data from Australia with some published data from Northern hemisphere.
  2. Several statements, in particular, the conclusions are not adequate. These should be significantly revised.

Author Response

Reviewer 3

The authors investigated CVD hospitalisations and deaths across seasons and during the December holiday period using retrospective, longitudinal, population-based cohort data from Queensland, Australia (from January 2010 to December 2015). Overall, it's an interesting study with strong clinical relevance and significance. I only have a few comments.

  1. The authors should compare their data from Australia with some published data from Northern hemisphere.

Response: Thanks for this comment: We have compared our finding to published works in the northern hemisphere on 211-219 on page 7.

  1. Several statements, in particular, the conclusions are not adequate. These should be significantly revised.

Response: We have created a new section for the conclusion and have revised it accordingly on page 8.